# A Qualitative Evaluation of the Barriers and Enablers for Implementation of an Asymptomatic SARS-CoV-2 Testing Service at the University of Nottingham: A Multi-Site Higher Education Setting in England

**DOI:** 10.3390/ijerph192013140

**Published:** 2022-10-12

**Authors:** Holly Blake, Sarah Somerset, Ikra Mahmood, Neelam Mahmood, Jessica Corner, Jonathan K. Ball, Chris Denning

**Affiliations:** 1School of Health Sciences, University of Nottingham, Nottingham NG7 2HA, UK; 2NIHR Nottingham Biomedical Research Centre, Nottingham NG7 2UH, UK; 3School of Medicine, University of Nottingham, Nottingham NG7 2RD, UK; 4Executive Office, University of Nottingham, Nottingham NG7 2RD, UK; 5School of Life Sciences, University of Nottingham, Nottingham NG7 2UH, UK; 6Biodiscovery Institute, University of Nottingham, Nottingham NG7 2RD, UK

**Keywords:** qualitative, SARS-CoV-2, COVID-19, health protection, health testing, pandemic

## Abstract

Asymptomatic testing for SARS-CoV-2 RNA has been used to prevent and manage COVID-19 outbreaks in university settings, but few studies have explored their implementation. The aim of the study was to evaluate how an accredited asymptomatic SARS-CoV-2 testing service (ATS) was implemented at the University of Nottingham, a multi-campus university in England, to identify barriers and enablers of implementation and to draw out lessons for implementing pandemic response initiatives in higher education settings. A qualitative interview study was conducted with 25 ATS personnel between May and July 2022. Interviews were conducted online, audio-recorded, and transcribed. Participants were asked about their experience of the ATS, barriers and enablers of implementation. Transcripts were thematically analysed. There were four overarching themes: (1) social responsibility and innovation, (2) when, how and why people accessed testing, (3) impact of the ATS on the spread of COVID-19, and (4) lessons learned for the future. In establishing the service, the institution was seen to be valuing its community and socially responsible. The service was viewed to be broadly successful as a COVID-19 mitigation approach. Challenges to service implementation were the rapidly changing pandemic situation and government advice, delays in service accreditation and rollout to staff, ambivalence towards testing and isolating in the target population, and an inability to provide follow-up support for positive cases within the service. Facilitators included service visibility, reduction in organisational bureaucracy and red tape, inclusive leadership, collaborative working with regular feedback on service status, flexibility in service delivery approaches and simplicity of saliva testing. The ATS instilled a perception of early ‘return to normality’ and impacted positively on staff feelings of safety and wellbeing, with wider benefits for healthcare services and local communities. In conclusion, we identified common themes that have facilitated or hindered the implementation of a SARS-CoV-2 testing service at a university in England. Lessons learned from ATS implementation will inform future pandemic response interventions in higher education settings.

## 1. Introduction

COVID-19 is a major global public health burden, associated with high morbidity and mortality [1]. In a higher education context, the negative impacts of COVID-19 and social restrictions on students and faculty are well-documented [2,3,4,5,6]. On university and college campuses, virus transmission risk is high, associated with a social culture [7]. During the COVID-19 pandemic, regular surveillance testing of asymptomatic individuals for severe acute respiratory syndrome coronavirus 2 (SARS-CoV-2; a strain of coronavirus that causes COVID-19) has been adopted as one approach to identifying and managing virus outbreaks in higher education settings [8]. Internationally, regular testing regimens have been shown to be cost-effective, reduce infection rates and increase confidence in safety among staff and students in other educational settings (i.e., kindergarten to 12th grade (K-12) schools in the United States) [9,10].

Evaluations of testing programmes conducted on university settings commonly focus on test performance [11], number of tests performed, disease transmission and infections detected, or the technical, logistical, and regulatory processes that enabled the scale-up of testing onsite [12,13,14,15,16,17,18,19,20,21,22,23,24,25]. Some studies include surveys which are primarily focused on student transmissibility or symptom reporting [23,26].

There are few evaluations of the implementation of SARS-CoV-2 testing programmes or services in universities, from the perspectives of service users or service delivery personnel. Some studies have investigated students’ views towards the service, or established testing acceptability among service users [27,28,29]. A recent study explored the facilitators and barriers to the implementation of SARS-CoV-2 testing in a university setting [30]. This was a mixed-methods process evaluation of the University of Nottingham asymptomatic testing service (ATS), investigating adherence to a specific residence-based testing protocol and the views of students and staff towards the service offer, after 7–8 months of service operation. Data were collected during the national escalation of the Delta variant in England in spring 2021. This service continued to deliver testing over an extended period (two years: September 2020–July 2022), through multiple national surges of COVID-19, and the emergence of several new variants. The aim of the current study is to explore the views of service personnel towards the implementation of a SARS-CoV-2 testing service over a sustained period during a pandemic. The research questions (RQs) were: (RQ1) What are staff views towards the availability of the ATS? (RQ2) What are the barriers and enablers of service implementation? (RQ3) What was the perceived impact of the ATS on the spread of COVID-19? (RQ4) What lessons can we learn from ATS implementation to inform future pandemic responses in universities?

## 2. Materials and Methods

This was a qualitative study using semi-structured interviews [31,32] with university employees involved in the delivery of the University of Nottingham ATS. The study was approved by the University of Nottingham Faculty of Medicine and Health Science Research Ethics Committee (REC reference: FMHS 96-0920). We used an interview topic guide, consisting of key open-ended questions and probes, to guide our interviews (Appendix A). This was developed by the lead author (health psychologist and health services researcher), based on review of relevant literature, and reviewed by the research team (health researchers), service lead, and two service users (university employees). The interview guide was pilot tested and minor edits were made to improve flow and clarity.

The sample was identified and recruited through the service administrator. Potential participants received an information sheet stating the research aims and could take part by emailing the researchers to state their interest. The information sheet informed them that they could withdraw from the study at any time without giving reason. All participants gave written consent to take part via an online consent form hosted on Jotform survey software. We adopted a convenience sampling approach and gave the opportunity to participate to all individuals who had undertaken a role within, or associated with, the ATS and were available for interview during the study timeframe. The intention was to interview people with either strategic or operational roles in service implementation and gather insights from staff with involvement in diverse aspects of the ATS (e.g., service administration or management, technical or laboratory staff, and research staff), including those at the early stages of their careers, through to senior leaders.

Interviews were conducted between May and July 2022. For national and local context, this follows a period of dominance of the Omicron BA.2 variant in the UK from January to May 2022 and the emergence of several recombinant variants (e.g., XF; a recombinant of Delta, XE; a recombinant of Omicron BA.1 and BA.2). Omicron BA.4 and Omicron BA.5 were declared Variants of Concern in May 2022 and Omicron BA.5 became more prevalent, and subsequently, the dominant SARS-CoV-2 variant by August 2022. Concurrently, the emergence of variant BA.2.75 resulted in a small number of cases identified in the UK in July 2022 [33]. Over a two-year period, the University’s ATS tested over 150,000 samples for SARS-CoV-2 RNA, identifying almost 2000 positive cases in staff and students who displayed no symptoms of COVID-19 at the time of testing. Further details about service provision from pilot stage to rollout of the full service are provided elsewhere [28,30]. Details of the ATS delivery and timeline, within the changing pandemic context, are provided in Figure 1. The methods and validity of the testing approach have been published elsewhere [34,35,36,37].

Interview data were collected using a video-conferencing platform, were audio-recorded with consent and fully transcribed. Field notes were taken during interviews. Researchers who conducted the interviews (IM, NM) and the researcher leading the data analysis (SS) had no prior relationship with any of the participants and were not involved in any aspect of service delivery. All researchers had a background in health disciplines and were trained in interview skills and good clinical practice. No incentives were offered for participation. Interview length varied from 24 to 80 min (mean: 44 min).

We adopted thematic analysis as our analytic approach, informed by Braun and Clarke’s six-phase guide for inductive thematic analysis [38]. Analysis of data was undertaken in tandem with data collection, using NVivo (Release 1.0). Coding and analysis were led by an experienced qualitative researcher (SS) who familiarised themselves with the data through repeated readings of transcripts and note-taking. SS generated initial codes which were refined and added to as interviews were revisited over time. IM and NM independently analysed a subset of 30% of the transcripts, then reviewed and cross-validated the coding for the whole dataset. All three researchers (SS, IM, NM) and the lead author (HB) discussed and reflected on codes which informed the ongoing coding of the data. Once coding was complete, SS developed larger patterns across the dataset and grouped the codes into potential themes [38,39]. Related clusters of coded text were then grouped together conceptually to form subthemes under the main themes to describe the interpretation of the data. Themes and subthemes were then reviewed by IM and MN and discussed between the three researchers, to check they captured the essence of the coded extracts and dataset. The researchers worked together to refine the themes through investigator triangulation [39]. The final analysis reported is based on the combined interpretation of the data. Respondent verification was undertaken to strengthen trustworthiness; one participant listened to nine (36%) of the interview recordings and checked transcripts for accuracy. To focus this paper, we have chosen to analyse data from question items related to views towards the service and the barriers and enablers of service implementation. Data on specific workforce and career impacts of service involvement were collected in the same interviews but are explored in depth elsewhere.

## 3. Results

In total, 58 employees were invited to participate in an interview. Of these, 25 participated (men: *n* = 8, 32%; women *n* = 17, 68%) in the interviews, 1 declined, and 32 did not respond. Interview participants were aged 26 to 63 years (mean = 40.72, s.d. = 12.64). We did not collect reasons for non-participation, although those who offered reasons for decline commonly reported competing demands on their time. Participant characteristics are provided in Appendix A. The analysis produced four overarching themes and nine sub-themes, shown in Figure 2.

### 3.1. Theme 1: Social Responsibility and Innovation

#### 3.1.1. Sub-Theme 1: Positive Impact on Community

In establishing the service, the institution was seen to be socially responsible, valuing its staff and students, and the general public through actions to prevent onwards transmission. Participants felt that the availability of the testing at the institution sent a broader message that it cares about its staff, students, and the local community. The visibility of the ATS within and outside of the institution showed that the University was taking the pandemic seriously.


*“…whenever we started the testing site momentum, that had a very strong and positive impact on the community …by us taking the testing out to the community area and being visible it showed how serious we took the pandemic itself …just being more visible was pretty big.”*
(Respondent 5)

#### 3.1.2. Sub-Theme 2: Impact on Safety and Wellbeing

For University staff the availability of the testing had a positive impact on wellbeing, particularly for those feeling pressure to return to face-to-face teaching. The availability of testing for their students helped to alleviate some of the worry associated with returning to campus and the potential for increased exposure to the virus.


*“…having these sorts of testing things would mean that if they were in, they knew that most of the people there were being tested and were therefore safe…I think it was incredibly reassuring.”*
(Respondent 12)

There was a sense that the availability of the testing helped people to feel safe, particularly those who were at increased risk from COVID-19, or those with vulnerable family members. The testing offered peace of mind that might not otherwise have been achieved.


*“They were incredibly grateful for it because a number of members of staff had ‘at risk’ partners for health reasons to covid so to know that they could test regularly and know that they were not going to put their partner or family member at risk was a huge benefit for them.”*
(Respondent 15)


*“…they’ve been getting tested to make sure that they, they keep their, their partner safe and therefore they, the, the wider population safe as well.”*
(Respondent 8)

Some participants suggested that the ATS was perceived to be of greater value by staff than students; this belief was associated with a perception that becoming infected with the virus would have greater impacts for staff (compared with most students) in terms of family life and work commitments.


*“…Staff would have been more worried because the isolating for 10 days would have meant losing 10 days of work, or potentially bringing COVID to their family, kids etc and I think, if anything it was the value of the service was appreciated more by staff in my opinion for these reasons.”*
(Respondent 3)

### 3.2. Theme 2: When, How and Why People Accessed the Testing

#### 3.2.1. Sub-Theme: Barriers and Enablers of Testing Uptake

Although the service was consistently utilised, there were some challenges with encouraging regular testing, particularly among students. This was evidenced by a notable variation in testing uptake observed by ATS staff at different times of the year. ATS staff reported a surge in testing at critical times, for example, around the departure of most students from university campus for the winter break. This was associated with students’ desire to protect friends and family members back at home.


*“…the sheer demand of the test at peak times like end of terms or before holidays, like Christmas, they were very popular times for the test, people heavily relied on the test to feel safe, to go home to their family …”*
(Respondent 3)

An incentives scheme was introduced to facilitate the uptake of testing with the ATS, based on a trial-and-error approach. While some incentives were more effective than others at encouraging testing uptake, the scheme was perceived to work well, with reports of increased uptake by students and in some cases, staff.


*“we always saw an uptake in the number of students who were testing when, when it was incentivised.”*
(Respondent 24)


*“… it’s a great window into human psyche that offering a relatively small treat as it were, gave people a positive reason to, uh, to, to want to donate a, a saliva sample rather than just being told that they had to…”*
(Respondent 8)

ATS staff felt that the type of test on offer helped to encourage testing uptake. At the time of the COVID-19 pandemic, the ATS was offering a novel method of testing for SARS-CoV-2 RNA. The testing involved the provision of a saliva sample in a tube which was then taken and analysed in the lab. Many participants saw the saliva test as more comfortable to administer than the available alternatives (e.g., nasal swab, throat swab).


*“…they just simply didn’t like the swab testing…the ability to just basically dribble into a tube and then pop that in a bag and leave it somewhere for somebody to collect and then report back whether you’re positive or not positive…seemed very beneficial.”*
(Respondent 8)

The testing was free and readily available onsite to maximise uptake. It was provided in central locations with heavy footfall, student residences, and offices: “*…the accessibility of the testing, and the fact that, there were so many locations where... it could, you know, you could drop off a sample, whenever it suited you.*” (Respondent 1). While this facilitated access to testing, particularly for students, its impact on uptake was unclear. It was proposed that some students may simply not engage in testing irrespective of convenience or location, as they would be required to isolate if they received a positive result.


*“…the students were so worried that they would test positive, and then the whole floor [of the residence] would need to isolate…human nature was different to what we predicted.”*
(Respondent 13)

ATS staff identified the accessibility of testing, free tests, and the availability of tests for family members as key facilitators for uptake of testing. Participants perceived the ATS to be responsive to a rapidly changing external context; as government guidance changed and children were required to have a COVID-19 test to access school, the service expanded to accommodate this.


*“… we extended our provision to cover primary school age children of staff to help them when the schools were requiring regular test …I do believe that the staff really value the service that we provide.”*
(Respondent 16)

However, the rapidly changing government guidance led to some confusion among staff and students around when, and how, people should test. This was perceived to be one of the greatest barriers to the implementation of the testing service. For example, when national social restrictions were relaxed, it then became more difficult to engage students with the testing service even when incentives were used.


*“…the easing of … the rules to a way that negatively impacts on the people’s desire to test.”*
(Respondent 7)


*“…when the national regulations were eased off and pretty much scrapped it meant we got less people testing… our sample numbers fell through the floor…government guidelines definitely hindered the sort of, the input and output of the service.”*
(Respondent 10)

Participants observed some variation in the uptake of testing between student groups. They reported that undergraduate students appeared to access the testing less frequently than postgraduate students: “*I think maybe postgrads were more keen also because you know some of them are a bit older…*” (Respondent 2). There was higher testing frequency observed among students registered for healthcare degrees, and this was associated with the requirements of their course of study (i.e., testing being a requirement to attend placements): *“…students who were on placements so, especially medics and nursing, midwifery, those students tested regularly with us as did the vets.”* (Respondent, 15). A summary of ATS staff views on the factors that helped or hindered uptake of testing is shown in Figure 3.

#### 3.2.2. Return to Normality

The provision of the ATS testing service enabled *“return to relative normality earlier than other people”* (Respondent 12). The availability of the service also enabled students and staff to test daily, and within a much shorter time frame than the alternatives available at the time of the pandemic.


*“…that thing with the NHS, which was the alternative…you could order lateral flows, or you could ask for PCR [polymerase chain reaction], but the PCR you were supposed to have symptoms to get it but with our PCR test, they could be testing daily, even without having the symptoms so they could catch the infection extremely early on.”*
(Respondent 2)

### 3.3. Impact of the Service on the Spread of COVID-19

#### 3.3.1. Identifying and Containing Outbreaks

The launch of the ATS coincided with a national increase in positive cases of COVID-19, and therefore, a changing national context. Although the ATS was perceived to be positive action taken by the University to identify and manage outbreaks of COVID-19, most staff on the ATS reported that the service had minimal impact on spread of the virus at the outset. Nevertheless, it was generally thought that the ATS was successful in highlighting where outbreaks were taking place at that time, although there were some concerns relating to how positive cases should be managed and advised on an individual basis, given frequently changing, and often ambiguous, government advice at the time.


*“I think we ended up just chasing outbreaks rather than preventing them, which was fine because we just identified lots of problems, but we didn’t know what to do with the outbreaks once they were there other than to say isolate but because the government guidance changed so much, you know Halls are not a household, household isn’t a massive building so how you know, it’s almost like we needed some clearer guidance on what to tell students once they were positive.”*
(Respondent 1)

ATS staff felt that after the initial surge of COVID-19 in autumn 2020, the service was able to help contain outbreaks of COVID-19 within the student population, and that this had a knock-on effect in the wider community by preventing onwards transmission. However, how effective the testing could be at containing COVID-19 outbreaks depended heavily on individuals with a positive test observing an isolation period. Self-isolation was recognised as a situation that some students were keen to avoid, particularly in the early days of the pandemic: *“I think there were a couple of instances where a few students were are not happy about the isolation that came with positive results”.* (Respondent 9). Given the challenges of ensuring that students engaged with the testing during national surges of COVID-19, some ATS staff felt that testing should have been mandated at that time, to contain outbreaks within the University population.


*“…if testing was mandated, it would be a more effective service, and a large amount of time and thought was given to how we engage with students, where it’s an optional process.”*
(Respondent 24)


*“I don’t think it was helpful in stopping spread, because of, the inability of the university to make it a requirement…I was met with, well, they can’t mandate testing…”*
(Respondent 21)

The service overall was viewed as an asset to the university. The staff of the ATS felt the service helped to contain COVID-19 outbreaks through the pandemic, but they could not always fully prepare for surges on the university campuses. The ATS was seen to have significant impact beyond the university; staff from the ATS were able to offer testing assistance to local NHS services during peak times. It was perceived that, without the support from these extra staff from the University, local outbreaks in the community might have either continued for longer or become more widespread in the general population.

#### 3.3.2. Relationship with Social Behaviours

One barrier to containing COVID-19 outbreaks was the perceived inability of the ATS to influence social behaviours. There was a general feeling that: “*…you can lead a horse to water, but you can’t make it drink*”. (Respondent 6). Those accessing testing regularly were viewed to be individuals who would be more likely to be following government guidance in other areas (e.g., testing, social distancing, hand hygiene, face coverings) to reduce the spread of COVID-19.


*“I think the people who were going to follow the rules and social distance and wash their hands …they would have done so even if the Testing Service wasn’t in place, but it might have been a good reminder.”*
(Respondent 21)

ATS staff reported that people who accessed the testing had a mixed response to their test results. Some students and staff receiving a negative test result would minimise contact with others and follow the government guidance at that time (i.e., social distancing). For others, the relief of receiving a negative test result facilitated social behaviours that were seen as “taboo” at the time, such as hugging and close contact.


*“…people went to the either extreme…you would have people who were like ‘I am negative, so give me a hug, so it’s fine, we will share a drink’ or whatever and then you had other people who were the opposite who said ‘I’m negative, I don’t’ want to get it, so don’t go near me.’ But it forced people into two very extreme views and behaviours, I think. I don’t think anyone was really in the middle by the end.”*
(Respondent 1)

### 3.4. Lessons Learned for the Future

#### 3.4.1. Service Offer

Those working in the ATS offered suggestions for improving the service offer. Many interviewees felt that the service needed to offer the entire package from the initial testing through to support for isolation.


*“…but I guess that what the service is a testing service, you know if it wanted to also, it could have extended to isolation service and wellbeing, but we gave that back to the Uni to do really. It was like at the end of our remit, erm, which you know, was hard because we were the ones telling people, and ‘cause it was a telephone call, it wasn’t just an email, that you didn’t know these people, you got to hear their voices, got to hear how they reacted, it was quite hard, because you want to do more…”*
(Respondent 1)

At the outset, the ATS only offered testing to students at the University, and this was later extended to staff and their families. In the view of ATS staff, this was an oversight; many expressed views that the testing should have been opened to university staff and their families sooner. When the service was opened to staff, the process to do so was slow and this was a barrier to uptake of staff testing.


*“…I was pushing in those early days that we had opened up the testing service much more quickly to, to staff members … better integration of, of the students and the staff early on, and then expanding it out more rapidly to the, the family members.”*
(Respondent 8)

#### 3.4.2. Service Delivery

Regarding service delivery, participants felt that the planning and rolling out of the ATS service could have been quicker. While the service itself was visionary and innovative, some ATS staff perceived that there was not enough forward thinking in preparation for the return of students to campus in September 2020, and that more work could have been done over the summer to ensure the testing service was up and running prior to student return to campus: “*By the summer it was obvious this wasn’t something that was going to go away.*” (Female, 26).

Several ATS staff felt that the service did not have enough authority to deal with the behavioural consequences of testing. Following government guidance, the ATS was unable to enforce isolation for those testing positive for COVID-19 and that this in turn created challenges in containing virus outbreaks.

A small number of participants felt that there was a missed opportunity to roll the ATS out at a national level. This was associated with a lack of knowledge on how to achieve policy change, or a lack of essential contacts within a “*it’s who you know*” culture at government policy level. There appeared to be unmet training needs among staff working towards service accreditation regarding the appropriate mechanisms and platforms for influencing policy.


*“…a bit of a regret there that we weren’t able to get better engagement with the government departments to roll out what arguably is a simpler and an easier test.”*
(Respondent 8)

#### 3.4.3. Ways of Working

Given the rapid set-up and deployment of the ATS, the nature of the service provision involving development of new technology, and the establishment of new teams at pace, participants shared lessons to be carried forward. The prevailing view was that the ATS has created a footprint for more collaborative ways of working: “*I think it’s spearheaded an inclusive approach for collaboration across professional services and faculties.*” (Respondent 15). It was perceived that, prior to the pandemic, the organisation was “*incredibly risk averse*”, with organisational bureaucracy and “*institutional barriers*” which hindered or prevented action or decision-making. The structural changes afforded by the organisational response to COVID-19 and reduction in formalities and administration, was seen to increase efficiency.


*“…throw money and give us roots to get problems solved without a huge amount of red tape…we can just operate to get something done...having a hotline to the people that can actually make decisions that the university will follow was particularly welcome.”*
(Respondent 12)

This allowed the ATS team to respond rapidly to changes in the external environment that impacted on service offer or delivery: “*We changed and adapted so quickly to new guidelines.*” (Respondent 16).

A key facilitator to the success of the service was a positive, inclusive leadership team, who were perceived to be approachable and: “*created a very special atmosphere where it is 100% two-way communication.*” (Respondent 5). Participants valued the regular communication with updates on service status and arising issues (e.g., low testing uptake, logistic problems, changes to government guidance impacting on communications with staff or students); this facilitated team problem-solving and in turn, service responsiveness given the changing external context.

## 4. Discussion

To the best of our knowledge, this paper reports the first qualitative study focused on the views of service operations personnel towards the rapid establishment and implementation of an asymptomatic SARS-CoV-2 testing service in a higher education setting, through the changing but sustained context of the COVID-19 pandemic. The findings of this study, conducted at a multi-site university in England, have relevance for health protection initiatives in educational settings worldwide.

The research questions (RQs) generated data that are discussed in line with the four over-arching themes: (1) social responsibility and innovation, (2) when, how and why people accessed testing, (3) impact of the ATS on the spread of COVID-19, (4) lessons learned for the future. The discussion of themes 1–3 maps to RQ1–3. Theme 4 maps directly to RQ4, which is cross-cutting (across all themes 1–4).

### 4.1. Theme 1: Social Responsibility and Innovation

We explored staff views towards the availability of the ATS (RQ1), which pointed to the perceived positive impact of the service on community, safety, and wellbeing. This theme highlights the importance of the organisational context as a barrier or enabler of service set-up and implementation (RQ2). Published views towards mass asymptomatic testing in university settings have been mixed [40,41,42], although our participants were largely positive about, and took pride in, the ATS and what it set out to achieve. Our findings show that by establishing the ATS as a pandemic response, the institution was perceived by the ATS workforce to be socially responsible, demonstrating value for its community of staff and students, and the wider public. Universities must contribute to addressing urgent global social and environmental challenges; the COVID-19 pandemic offers ample opportunities for organisations to actively engage with corporate social responsibility strategies and agendas [43,44]. In our study, the ATS was recognised as an act of social responsibility in extraordinary times. Social responsibility is aligned with university values and is purported to foster empathy in organisations and mobilise leaders in the creation and implementation of new practices [45]. The operation of a SARS-CoV-2 testing service was seen to protect the university population and extend wider societal benefit through provision of testing support for partner organisations to achieve mutually beneficial goals associated with containing the spread of COVID-19 (e.g., working in partnership with, and supporting the NHS, local government, Public Health England—“we are all in it together”). This was enhanced by high visibility of the service within and outside the university, a marker of authenticity, exposing social impact through visible actions and impact.

Within the university, implementation of the ATS was facilitated by the internal context; primarily, a reduction in bureaucracy and red tape that allowed decisions to be made, and processes implemented, at pace. Examples include streamlined approvals processes, direct and visible support from senior leaders, ‘flattening of the hierarchy’, inclusive and collaborative teamworking and clear channels of communication. This had important implications for the pandemic response at this institution. Red tape is negatively associated with proactive behaviour [46]. Proactive behaviour is future-focused, change-oriented, and associated with innovation [47]. It plays a key role in the effectiveness of organisational responses to changing conditions [48], such as the COVID-19 pandemic. Within the ATS, transformational leadership (e.g., influencing, transforming, and inspiring others), a distributed leadership style (e.g., where teams share responsibility and work towards a common goal) and proactive individual and team behaviour facilitated the implementation of the service. The impact of university investment in employee development through the ATS is evident in our data and interrogated elsewhere [49].

The ATS was the first accredited asymptomatic testing service in a university setting in the UK (one of only eight accredited SARS-CoV-2 testing laboratories at the time). There were some challenges experienced in engagement between ATS scientists and policy makers throughout the ATS accreditation and implementation, which led to delays and barriers to roll out and upscaling of the ATS novel testing approach. In part, this was perceived to be due to a lack of clear routes into policy from academia, with opaque decision-making and impenetrable institutions. Nonetheless, the different mentalities and imperatives of scientists and policy makers, and the challenges of establishing “mutualistic relationships” between these two populations are well recognised [50].

### 4.2. Themes 2 and 3: Testing Uptake and Service Impact on the Spread of COVID-19

Themes 2 and 3 are inter-linked, focusing on practical and behavioural aspects of testing uptake in the target university population (Theme 2: how, when and why people accessed testing), and the consequences of service delivery for outbreak prevention or management (Theme 3: perceptions of the impact of the service on the spread of COVID-19). This generated insights into the practical and logistical factors that enabled or hindered testing for students and staff (RQ2), and the perceptions and behavioural response of students and staff as the ATS was delivered (RQ3).

The negative impact of the pandemic on university staff and students has been recognised [2,3,4,5,6,51]. Our participants indicated that the provision of testing onsite was beneficial, both for wellbeing and a perception of feeling safe on campus. Our findings align with those of a previous study which revealed a strong appetite for widespread testing across all campuses to maximise perceptions of safety in the student and staff body, and positive impacts on wellbeing (for staff and students) that were directly associated with the provision of virus testing at the same university [28].

In a pandemic context, there is a need to act in situations of high ambiguity and uncertainty, often using trial-and-error strategies [52]. In this context, staff viewed the ATS to be flexible and highly adaptive over time. It was broadly successful in identifying outbreaks, and to some extent managing them, although the latter was challenged by factors external to the ATS (e.g., fluctuating, conflicting or ambiguous national messaging and the behavioural consequences, low personal risk perception in the student population, signposting elsewhere for follow-up support for students needing to isolate). Others have shown that regular surveillance testing, alongside other measures (e.g., isolation, contact tracing, quarantine) offers an effective means of identifying breakthrough infections, halting onward transmission, and reducing total caseload [22,53]. In this service, gaining consistency in testing uptake was difficult, and others have also reported that achieving high levels of participation in university settings is challenging [31]. Incentives were used to maximise participation, although the effectiveness of incentives was variable. Simple incentives, such as free food or drink for those who were tested, was seen to encourage testing uptake in students. Financial incentives have been used with some success to promote uptake of other types of health testing in university settings (e.g., sexual health screening [54]) and in studies of university-based COVID-19 surveillance testing (to nudge testing uptake and daily questionnaire-based symptom and exposure reporting) [23]. However, monetary incentives may not be feasible to increase frequent testing uptake at scale, and over a long period, and staff/student views towards incentivisation are mixed [32]. In our study, the flexibility, convenience, and accessibility of testing, coupled with the simplicity of saliva testing were viewed by ATS staff to be key facilitators of testing uptake, as found in previous research [3,28]. Others have also found that convenience is necessary to ensure engagement with COVID-19 testing programmes [55].

ATS staff indicated that healthcare students were frequent testers, which likely reflects the requirement at the time that students undergo the same regular testing as other patient or client-facing staff in NHS and social care placements. Postgraduate students were more likely to be frequent testers than undergraduates, perhaps since they are more likely to be mature-aged, with family or work responsibilities [56]. Young adult undergraduates may perceive themselves to be at lower risk from COVID-19, since greater perceived risk is associated with higher uptake of preventive behaviours [57]. Indeed, prior research has shown that students with lower perceived risk of contracting COVID-19 take fewer tests in asymptomatic testing programmes [32]. Furthermore, COVID-19 and testing ambivalence was observed by ATS personnel in some students. Prior research has noted that the more ambivalent people are toward the behavioural recommendations associated with the COVID-19 pandemic, the less they report following them [58].

Our participants observed that testing participation rates significantly varied through the year, with higher uptake at critical points, such as the mass departure of students from campus, for vacations. Increased uptake at these times was viewed positively; testing was strongly advocated at these times to reduce community spread of COVID-19, since travel associated with academic calendars may influence the evolution of COVID-19 case rates [59].

ATS staff proposed that some students did not engage in testing to avoid the possibility of self-isolation. Self-isolation has been particularly challenging for university students, as evidenced by studies exploring the practical, social, and psychological impacts of social restrictions in student samples at the same institution, across various stages of the pandemic [2,3,60]. Regarding testing uptake, it has previously been advocated that uptake, adherence and satisfaction with testing is likely to be influenced by the level of support provided to students who are self-isolating [28,61]. This concern was echoed nationally at the time; with regular COVID-19 testing recommended in England from 9 April 2021 [62], the public’s social media response to government testing recommendations highlighted a major concern about support for self-isolation [63]. At our institution, some initial ‘teething problems’ were evident in student support during self-isolation during the pilot phase of the service [28], which were quickly resolved. However, the ATS remit only included testing and provision of results; those required to self-isolate were signposted to other university services for welfare support and follow-up. ATS staff believed this led to a lack of continuity in student support—this has important implications for student experience during a pandemic, and potentially, mental wellbeing, if students with support needs ‘fall through the cracks’ in service provision.

While the ATS helped to identify virus outbreaks, service operations staff observed that the availability of testing had mixed effects on student and staff behaviours; for some, it provided them with reassurance to engage in already permitted activities, whereas for others, it encouraged increased socialising and physical contact. The variation in individual risk perception and behaviours was also reported by service users (students and staff) engaging in university-based asymptomatic testing programmes [61,64].

### 4.3. Theme 4: Lessons Learned for the Future

This theme highlighted the lessons that can be learned from ATS implementation (RQ4). Staff reported lessons learned, which build on cross-cutting insights from themes 1–3 (RQ1–3) in which we described the key benefits of the ATS and the barriers and enablers of service implementation. These are summarised in Figure 4, together with lessons learned from ATS delivery. These insights can inform future pandemic responses in universities and educational settings worldwide.

### 4.4. Limitations

Data were collected at a single university in England. Although this institution hosts students and staff on multiple campus sites, the barriers and enablers of asymptomatic testing service implementation may vary across institutions, and geographical regions (nationally, and internationally). Data were collected during a period of service de-commissioning, during an ongoing (albeit improving) pandemic situation, which impacted on job roles for many of the participants and may have influenced their views or capacity to take part. Time constraints did not allow for longitudinal research, or more extensive respondent verification.

## 5. Conclusions

The COVID-19 pandemic presented a significant challenge to higher education settings. With the global spread of SARS-CoV-2, the University of Nottingham ventured into unchartered territory with the rapid establishment of a mass asymptomatic testing service for students, staff and their families, and the provision of support to local public health stakeholders for broader societal benefit. The development and implementation of diagnostic assays, the accreditation of the service and delivery of testing across multiple campuses of a large university, for almost two years, was a major contribution to the sector’s pandemic response, informing decision-making and actions by institutions within and beyond the UK. In establishing the service, the institution was seen to be valuing its community and socially responsible; the service was viewed to be broadly successful as a COVID-19 mitigation approach, particularly for the identification of outbreaks. Challenges to service implementation were the rapidly changing pandemic situation, delays in service accreditation and rollout to staff, ambivalence towards the virus, testing and isolating in the target population, and an inability to provide follow-up support for positive cases within the service. Facilitators included the simplicity of saliva testing, service visibility, reduction in organisational bureaucracy and red tape, collaborative working with regular feedback on service status, inclusive leadership, and flexibility in service delivery approaches. These factors will benefit future service delivery, accelerating decision-making and action. Future services should target both staff and students from the outset, and ideally offer testing in combination with follow-up support within the same service. More work is required to facilitate staff in mechanisms for upscaling service models beyond individual institutions and influencing national and international policy at pace. Overall, the ATS instilled a perception of early ‘return to normality’ and impacted positively on staff feelings of safety and wellbeing, with wider benefits for healthcare services and local communities.

## Figures and Tables

**Figure 1 ijerph-19-13140-f001:**
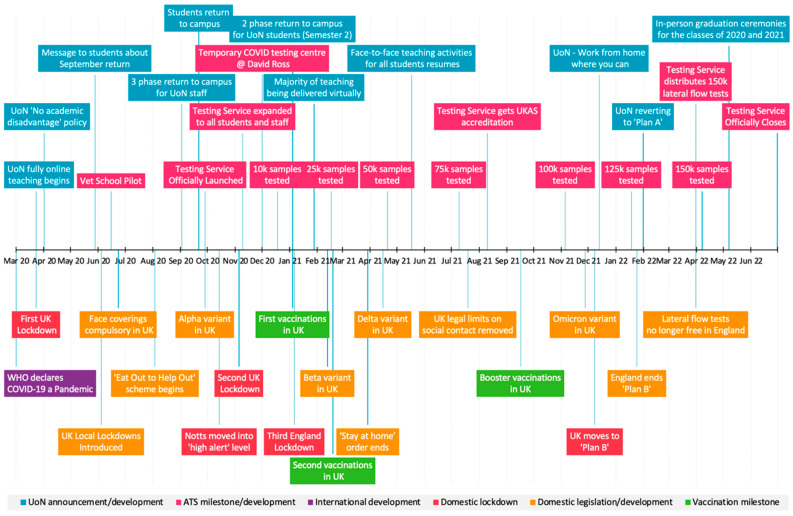
Asymptomatic testing service and COVID-19 timeline. UK: United Kingdom; WHO: World Health Organization; UoN: University of Nottingham; Notts: Nottingham; Vet School Pilot [19]; David Ross: University sports facility and temporary testing site venue; Plans A and B (see www.gov.uk (accessed on 8 October 2022)).

**Figure 2 ijerph-19-13140-f002:**
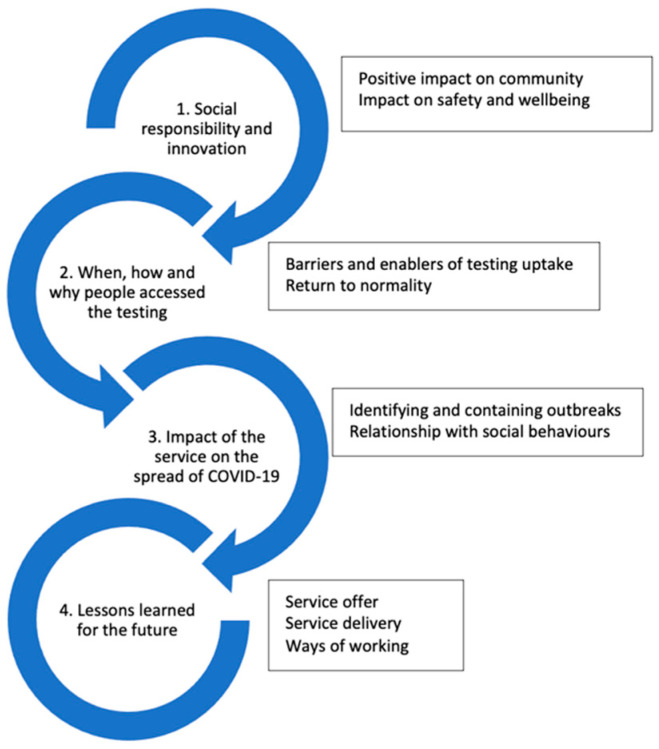
Overarching themes and sub-themes.

**Figure 3 ijerph-19-13140-f003:**
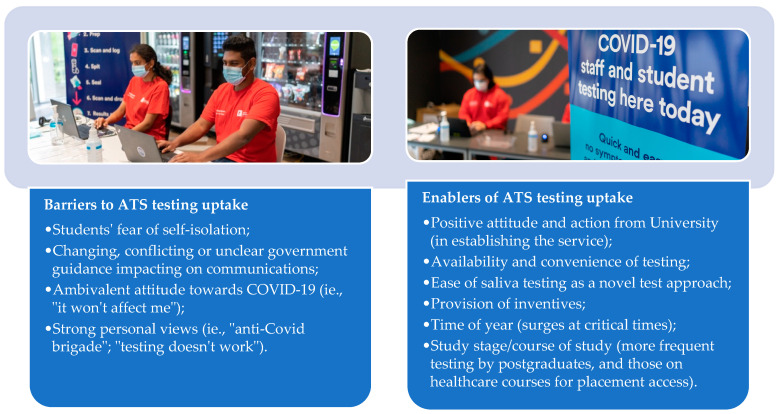
Barriers and enablers of ATS testing.

**Figure 4 ijerph-19-13140-f004:**
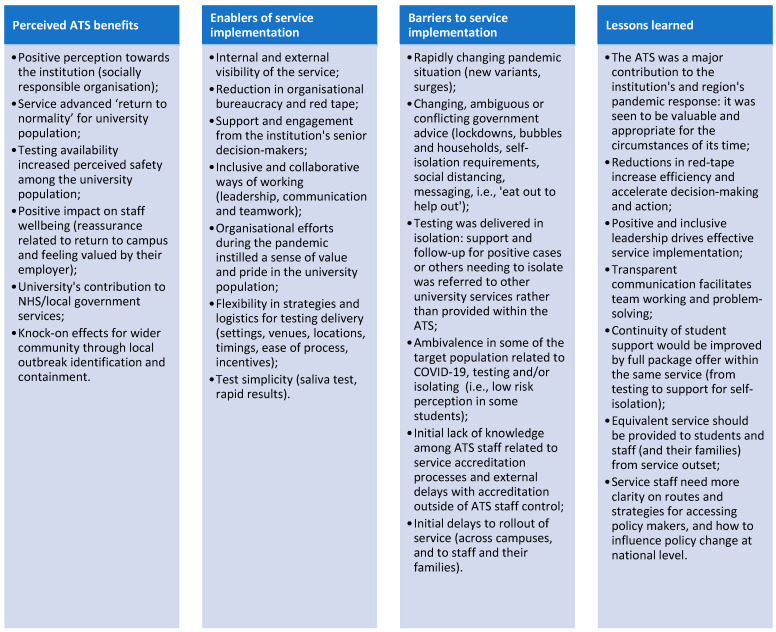
Perceived benefits, key enablers and barriers to service implementation, and lessons learned.

## Data Availability

The data that support the findings of this study are available on reasonable request from the corresponding author. The data are not publicly available due to their containing information that could compromise the privacy of research participants.

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
