# Peer review of "A Qualitative Evaluation of the Barriers and Enablers for Implementation of an Asymptomatic SARS-CoV-2 Testing Service at the University of Nottingham: A Multi-Site Higher Education Setting in England"

_ijerph, 2022, doi:10.3390/ijerph192013140_

Round 1
Reviewer 1 Report
With the risk of another wave of pandemic COVID-19 in the autumn, it becomes more important to identify factors for the widespread implementation of an asymptomatic SARS-CoV-2 testing service, which will help to reduce the scale of virus transmission - including in school settings. Such empirical studies involving staff providing asymptomatic SARS-CoV-2 testing services in school settings have not yet been conducted. This article therefore fills a diagnosed gap in the scientific literature.
The article is methodologically correct. The results of the study are correctly presented. In order to increase the quality of the article, the following corrections are recommended:
(1) modification of the title - the title is worded too broadly (the title is misleading the reader, the assessment of the barriers and enablers for implementation of an asymptomatic SARS-CoV-2 testing service in a UK higher education setting should not be made on the basis of research conducted in only one university, it is suggested to add in the title of the paper the narrowing "based on the example of research conducted in the University of Nottingham");
(2) completing the source base of the article (Authors should cite papers that outline the problem of COVID-19 screening at schools, for example, the research report COVID-19 Testing in K-12 Schools: Insights from Early Adopters, as well as articles published in the International Journal of Environmental Research and Public Health);
(3) the relatively small number of people covered by empirical studies raises concerns; however, as the research was qualitative in nature and, furthermore, all staff providing asymptomatic SARS-CoV-2 testing services in school were given the opportunity to participate, there is no need for change in this field; if the authors will continue this research in the future - it is recommended to increase the population of the surveyed people.
The above comments do not alter the high merit value of the article. It is recommended that the article be published with the above corrections (points 1-2).
Author Response
Please see attached letter.

Reviewer 2 Report
Title: A qualitative evaluation of the barriers and enablers for implementation of an asymptomatic SARS-CoV-2 testing service in a UK higher education setting
The manuscript “A qualitative evaluation of the barriers and enablers for implementation of an asymptomatic SARS-CoV-2 testing service in a UK higher education setting” identified common themes that have facilitated or hindered the implementation of a SARS-CoV-2 testing service at a university in England. It is well written article with some interesting findings; however, the article is providing a basic information about barriers and enablers for implementation of an asymptomatic SARS-CoV-2 testing in a UK higher education. Therefore, I would suggest to publish the data in a local journal, rather than publishing in an international journal. If authors can consider or provide the comparison data from other countries, then I would be better to consider the article in the current journal.
Author Response
Please see attached letter.

Reviewer 3 Report
Clarification is needed on the procedure and method, coding and on the codes and how they were made, as well as the categorizations.
The inclusion of a frequency table with the categorizations and the responses obtained is needed.
In addition, the validation processes should be included, e.g., the validation process:
collation and triangulation of data, Angulo Rasco (1990), saturation, auditors, contrasting of data, etc. See
(Quecedo and Castaño, 2003) and the existing literature on triangulation, some authors propose more specific strategies such as the "pairs test" (Goetz & LeCompte, 1988) in which three procedures to be followed are differentiated.
- Discussion section. It should be restructured following the parameters requested in the method and include in the conclusions the innovation practical and political implications to make the impact of your work clearer.
Author Response
Please see attached letter.
